# Integrating Force-based Manipulation Primitives with Deep Learning-based Visual Servoing for Robotic Assembly

Yee Sien Lee[1], Nghia Vuong[1], Nicholas Adrian[1,2], and Quang-Cuong Pham[1]

*Abstract*— This paper explores the idea of combining Deep Learning-based Visual Servoing and dynamic sequences of force-based Manipulation Primitives for robotic assembly tasks. Most current peg-in-hole algorithms assume the initial peg pose is already aligned within a minute deviation range before a tight-clearance insertion is attempted. With the integration of tactile and visual information, highly-accurate peg alignment before insertion can be achieved autonomously. In the alignment phase, the peg mounted on the end-effector can be aligned automatically from an initial pose with large displacement errors to an estimated insertion pose with errors lower than 1.5 mm in translation and 1.5° in rotation, all in one-shot Deep Learning-Based Visual Servoing estimation. A dynamic sequence of Manipulation Primitives will then be automatically generated via Reinforcement Learning to finish the last stage of insertion.

## I. INTRODUCTION

Robots have drastically increased industrial productivity by assisting humans to undertake high-volume and repetitive tasks such as lifting, assembly, and picking and placing of manufacturing parts. Specifically, robotic assembly has become progressively more common in the modern workspace, with increasingly complex and autonomous assembly tasks having been conducted in recent years [1]. However, robotic peg-in-hole assemblies require extremely high success rate and generalization to different contexts which are well beyond today's industrial robots' autonomous capability [2], [3]. Manual designing and fine-tuning are still required to achieve such tasks. Therefore, to achieve autonomous dexterous robotic peg-in-hole assembly, Vuong et al. proposed the idea of automatically discovering the dynamic sequence of Manipulation Primitives (MPs) via Reinforcement Learning (RL) [4].

The research from [4] utilized a force torque sensor to gauge the external force exerted on the end-effector. Besides maintaining a high accuracy, their method also showed promising generalization capability across different geometries. Nonetheless, the absence of a visual device limited the effectiveness of assembly as the peg had to be readily aligned within a small deviation range before insertion.

This study aims to improve the solely force-based solution of [4] in terms of practicality in real-world settings by implementing Deep Learning-based Visual Servoing (DLVS) in the alignment phase. In this project, the DLVS work by Yu et al. [5] was chosen to complement the solely force-based solution. With DLVS capability, the hole pose can be estimated automatically in the alignment phase.

[1]School of Mechanical and Aerospace Engineering, NTU, Singapore
[2]HP-NTU Digital Manufacturing Corporate Lab, Singapore

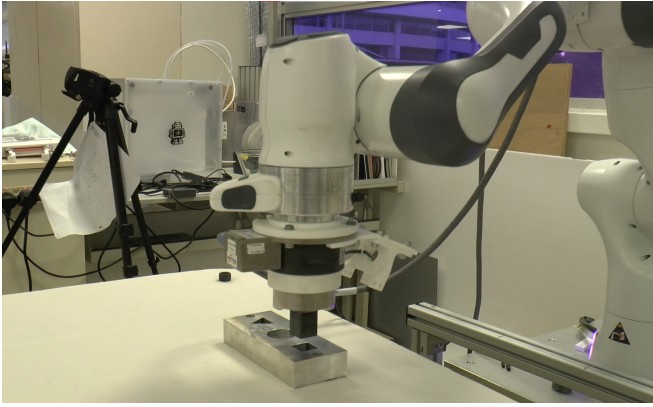

Fig. 1: Robotic assembly setup. A square peg was used in this study.

The scope of this study includes: (1) achieving high-accuracy ($< 1.5$ mm in translation and $< 1.5°$ in rotation) autonomous estimation of hole pose in the alignment phase, and (2) enhancing the generalization capabilities across workspace with the newly integrated DLVS feature.

## II. RELATED WORK

### A. Studies Achieving Alignment and Insertion

A study focusing on fast robust peg-in-hole insertion with continuous visual servoing was conducted in [6]. In the alignment phase, the peg was aligned to the hole based on heatmaps generated from a Deep Neural Network (DNN). After alignment, peg insertion was attempted via compliance using force-feedback. This approach was able to achieve high accuracy (peg-hole clearance of 0.015 mm). However, there were two downsides: (1) DNN in alignment phase could only align position, but not orientation; (2) In the insertion phase, a simple compliant force insertion which was unable to account for large rotational errors was applied.

[7] focused on achieving peg-in-hole assembly using multi-view images and DLVS trained on synthetic data. There were two steps in the alignment phase: (1) DLVS quickly moved the peg closer to the hole; (2) spiral search then precisely aligned the peg to the hole. The process would then proceed to the insertion phase where impedance control was used to perform the insertion. The clearance of the hole in this experiment was 0.4 mm. However, this approach could not align orientation errors as well due to the limitations of the DNN. Another downside was the long execution time. The approach needed more than 40 seconds to complete peg insertion from the start of the search phase.

### B. Alignment Phase: Deep Learning-based Visual Servoing

Deep learning-based visual servoing (DLVS) estimates the camera pose repeatedly while the robot is moving towards the target pose to achieve high final accuracy [5].

Bateux et al. explored an efficient method of generating dataset to train robust neural network for DLVS which considers changing lighting conditions and the addition of random occlusions [8]. The network achieves sub-millimeter accuracy but can only estimate a camera pose with respect to a fixed reference pose. The neural network has to be retrained every time a new reference pose is introduced, which is impractical in real-life usages. Thus, the authors proposed in the same paper another neural network which accepts a pair of images taken at random poses as input. Nevertheless, this extension could only achieve centimeter accuracy.

Yu et al. proposed a new neural network based on Siamese architecture that can output the relative pose between any pair of images taken at arbitrary poses with sub-millimeter accuracy [5]. The network is also effective under varying lighting conditions and with the inclusion of random occlusions, and can even generalize to objects with similar physical appearances. During actual insertion experiments, the model achieved sub-millimeter accuracy in camera pose estimation in one shot from initial deviations of: (-5, 5) mm for $x$ and $y$, (0, 10) mm for $z$, (-5, 5) deg for roll and pitch, (-10, 10) deg for yaw.

### C. Insertion Phase: Force-based Manipulation Primitives in Robotic Assembly

[3] proposed a robot manipulation technique that achieved faster cylindrical peg-in-hole insertions than human. Nonetheless, the author had to design manually the sequence of MPs, therefore the method did not have generalization capabilities to different contexts. Moreover, the sequence of MPs generated was fixed before execution. Thus, the approach suggested could not adjust to unpredictable circumstances in real-time execution. Some other works where a rigid sequence of MPs were manually defined include [2], [9], and [10].

In another study, Vuong et al. utilized RL to automatically discover the dynamic sequences of MPs [4]. The dynamically generated MPs could generalize across a wide range of assembly tasks and are more robust against environmental uncertainties during execution. Nonetheless, the method proposed was solely force-based. The lack of a visual sensor limited the effectiveness of robotic assembly tasks in real life due to the need for pre-defined peg pre-insertion alignment.

## III. METHODOLOGY

### A. Task Description

The peg-in-hole insertion task was split into two phases, namely (1) alignment phase and (2) insertion phase.

In the alignment phase, the expected outcome was the improved alignment between the peg and hole through the DLVS algorithm from [5]. The peg was then manipulated to move down until contact with the hole block. After rotating the resulting peg pose by $180°$ against $x$-axis and translating it down along z-axis by the hole depth, the estimated hole pose was recorded (Fig. 2). Before proceeding to the insertion phase, to check whether the alignment phase had achieved complete insertion, the peg was manipulated to move in $x, y, z$ - axes under two criteria, maximum movement duration and threshold of the force sensed. In unsuccessful attempts, the process subsequently proceeded to the insertion phase.

In the insertion phase, a dynamic sequence of MPs based on the RL policy trained in [4] were generated for final insertion. After each MP step, the policy would inspect the insertion status by finding the distance between the latest achieved pose and the estimated goal pose. If the $x$ and $y$ errors between the two poses were less than 5 mm and the $z$ errors were smaller than 4 mm simultaneously, the insertion would be deemed successful.

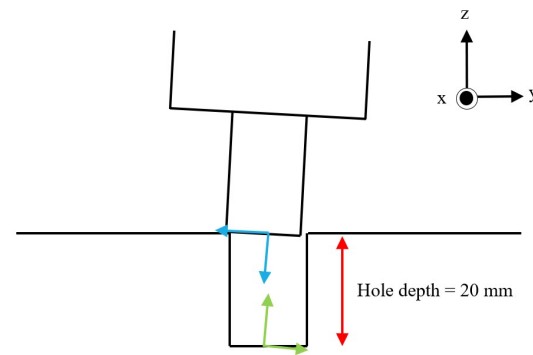

Fig. 2: Hole pose (green) estimation deduced from after-contact peg pose (blue) in alignment phase.

### B. Deep Learning-based Visual Servoing Neural Network

The neural network developed in [5] was designed to estimate the relative transformation between any two random camera poses. In training, the neural network took a pair of samples as input each time. Each sample in the pair comprised: (1) an image taken at a random pose and (2) the transformation matrix of the pose. The output of the network was the relative pose between the input pair of camera poses in the form of translation $(x, y, z)$ and quaternions $(a, b, c, d)$.

**Dataset Generation.** Firstly, the peg was guided manually to the insertion pose. The peg was then lifted vertically for 12 cm so that a full view of the target hole could be captured. This end-effector pose was then recorded as the default pose $T_d$ (Fig. 3). Samples were generated at random poses revolving around $T_d$. The origins of the new arbitrary end-effector poses were randomly sampled within a vertical cylinder of 10 mm radius and 20 mm height ($Cyl_{r=10, h=20}$), with the origin of $T_d$ at the the bottom center of the cylinder (Fig. 3). The rotation was randomly sampled within the range of $-10°$ to $10°$ for roll and pitch, and $-20°$ to $20°$ for yaw. At each random pose, an image was captured and the transformation matrix $T_{de}$ of the pose was recorded. $T_{de}$ is the transformation matrix which transforms the end-effector's coordinate frame to the default pose's coordinate frame. This

image and $T_{de}$ formed a complete sample which would later be input to the neural network as part of an input pair. The two input images in a pair were identified as $I_A$ and $I_B$. Before training, the true label which was the relative transformation $T_{BA}$ could be calculated as follows:

$$T_{BA} = [T_{dB}^{-1}][T_{dA}] \qquad (1)$$

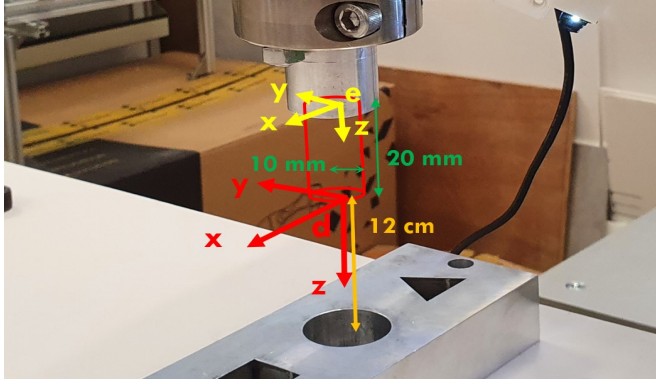

Fig. 3: The red coordinate frame defines the default pose, $T_d$. The origin of the new random end-effector pose, $T_{Oe}$ can be anywhere in the red cylinder.

As the robotic arm's shadows could affect the network's performance, the samples were generated with the hole being placed at different positions and orientations to ensure the shadows did not always appear at the same position. The hole was placed at 5 points on the base, where 4 points would form the vertices of a 5-cm square and 1 point would be at the center of the square. At each point, the hole block was rotated clockwise at $0°$, $30°$, $60°$, $90°$. At each orientation, 200 samples were collected. This would amount to 4000 samples (5 points $\times$ 4 orientations $\times$ 200 samples).

### C. Dynamic Sequences of Manipulation Primitives

Dynamic sequences of MPs could be discovered automatically through RL [4]. The RL policies were trained entirely in *Mujoco* simulation and transferred directly to physical execution.

**Manipulation Primitives in the Insertion Phase.** The MPs were defined as the appropriate motions of the end-effector in a task space. The motions were controlled by three types of instructions: (1) velocity command, (2) force command, and (3) stopping condition. The MPs were categorized into two families: free-space MPs and in-contact MPs. Free-space MPs were executed when the peg was not in contact with the hole block while in-contact MPs were executed when the peg was touching the hole block.

**Using Reinforcement Learning to automatically generate dynamic sequences of Manipulation Primitives.** The learning of dynamic sequences of MPs was regarded as a discounted episodic RL problem which could be addressed by a Markov Decision Process (MDP) [11]. An MDP is a function of state vector set $S$, action set $A$, state-transition probability $P$, reward $R$, and discount factor $\gamma$.

In this paper, an action $a$ was one of the MPs. The state vector $s$ was defined as the position of the peg relative to the estimated hole frame. After an MP had been executed at time $t-1$ and the stopping condition had been reached, the new state at time $t$ was measured. The reward function rewarded three terms: (1) MPs moving the peg closer to goal pose, (2) MPs with short execution time, and (3) MPs that had achieved SUCCESS stopping condition.

With an initial pose deviation of (-1.5, 1.5) mm and (-1.5, 1.5)°, this RL policy could achieve 94% success rate out of 50 insertion attempts with only one episode run. Thus, the peg's pose displacement errors needed to be within this range at the end of the alignment phase.

## IV. EXPERIMENTS AND RESULTS

The performance of the model was first evaluated on the test set. After that, the model was tested on actual insertion tasks. To prove the usefulness of the pre-insertion alignment, two baseline experiments were conducted. Lastly, the model was appraised for its generalization capability over workspace.

### A. Experimental setup

All experiments were conducted with a plastic square peg and a square hole which has 19.98-mm sides and 20-mm depth. The clearance between the mating parts was 0.26 mm.

The robot used in this project was the 7-DOF Franka Emika Panda cobot. An in-hand camera was mounted on the end-effector (Fig. 1). The camera was short-range with a field of view of $70°$ and a resolution of $640 \times 480$.

An additional force torque sensor, Gamma IP60 was used to measure the external force exerting on the peg as the force estimation in libfranka was too imprecise for the execution of force-based MPs.

### B. Training and model evaluation

Both samples in each input pair to the network had to be taken at random poses which were generated with respect to the same $T_d$. In total, there were $200^2 \times 20 = 800000$ pairs of samples taken from 20 sets (4 orientations at each of the 5 points). 80% of the samples were used for training and the rest were used as the test set. A model was trained for 10 epochs. The learning rate was $10^{-4}$ at the beginning and halved after the $4^{th}$, $6^{th}$, $8^{th}$ epoch. The batch size of the training set was 256. The entire training was run on 4 GTX-1080Ti's.

TABLE I: Test set errors ($e_\phi$: roll error, $e_\theta$: pitch error, $e_\psi$: yaw error).

| $e_x$ / mm | $e_y$ / mm | $e_z$ / mm | $e_\phi$ / ° | $e_\theta$ / ° | $e_\psi$ / ° |
|---|---|---|---|---|---|
| 0.2441 | 0.2875 | 0.2044 | 0.1792 | 0.1856 | 0.2148 |

The performance of the model on the test set is recorded in Table I. Since the RL policy used in the insertion phase could accept errors up to 1.5 mm in translation and $1.5°$

in rotation, the test errors were low enough to proceed to physical execution.

### C. Actual insertion task

The peg was manually guided to the goal pose at the beginning. 50 random poses within the sampling range defined by the $Cyl_{r=5,h=10}$ were generated around the goal pose. At each attempt, $I_A$ and one of the 50 images taken at the arbitrary poses, $I_B$ were input to the model. $\widehat{T}_{BA}$ between the two poses was estimated through DLVS. The peg would move to $\widehat{T}_{0A}$ at the end of the alignment phase.

In the insertion phase, the true hole pose was not given explicitly to the RL policy. The estimated hole pose deduced from $\widehat{T}_{0A}$ in the alignment phase was input to the policy. The peg was subsequently guided into the hole by a sequence of force-based MPs generated within one episode. The success rate and time taken are shown in Fig. 4.

### D. Comparing our method to baseline methods

Two baseline experiments were conducted to prove the usefulness of the proposed approach: (1) aligned peg with the same DLVS algorithm followed by pure compliance insertion and (2) attempted insertion with RL-generated MPs without alignment from the same sampling range defined by $Cyl_{r=5,h=10}$. As shown in Fig. 4a, both baseline methods' insertion success rates were much lower than that of our method whereas the time taken per attempt for alignment and insertion were much longer.

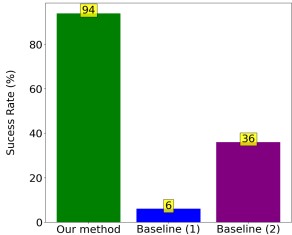
(a) Insertion success rates.

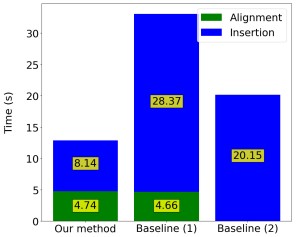
(b) Average time per attempt.

Fig. 4: (a) Insertion success rates and (b) Average time taken per attempt for alignment and insertion of our method compared to the two baseline methods out of 50 attempts. In (b), there was no alignment phase in baseline (2).

### E. Generalization over workspace

Iterative estimations with the same DLVS algorithm managed to align the peg to be within acceptable deviation threshold from initial pose differences that were larger than the sampling range $Cyl_{r=5,h=10}$. Two test cases (1 easy, 1 hard) were executed. In both cases, all pose errors converged to within 1.5 mm and $1.5°$ after a number of iterations (Fig. 5).

### V. CONCLUSIONS

The addition of DLVS has improved the practicality of the force-based peg insertion solution proposed by Vuong et al. [4]. With visual capabilities at the alignment phase, the

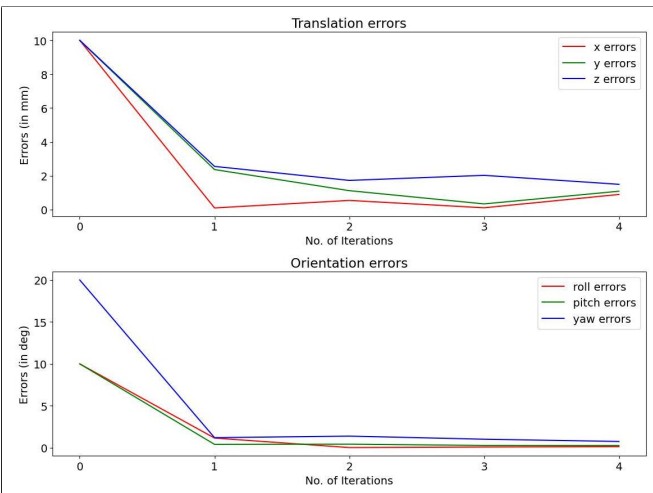
(a) Easy test case. Initial pose errors: (x, y, z) = (10, 10, 10) mm, (roll, pitch, yaw) = (10, 10, 20)°. Converged to acceptable error thresholds after 4 iterations.

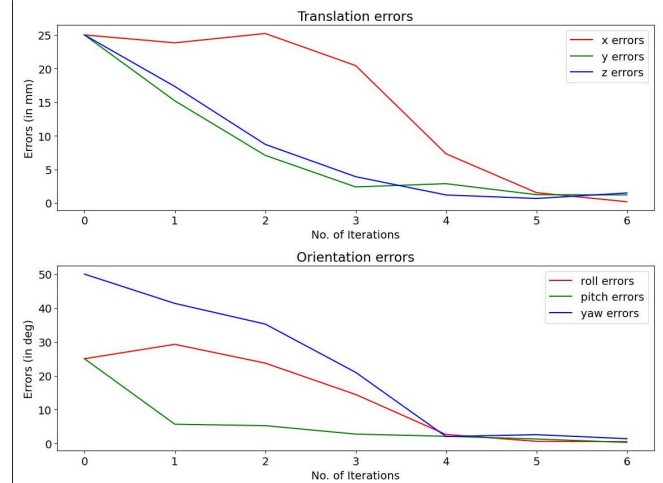
(b) Hard test case. Initial pose errors: (x, y, z) = (25, 25, 25) mm, (roll, pitch, yaw) = (25, 25, 50)°. Converged to acceptable error thresholds after 6 iterations.

Fig. 5: Initial pose differences that were larger than the sampling range $Cyl_{r=5,h=10}$ converged to within 1.5 mm and $1.5°$ in both easy and hard test cases.

peg's starting pose error thresholds in both translation and orientation were increased. Even initial pose differences that were larger than the normal sampling range could be handled if iterative visual servoing was applied.

Furthermore, the true hole pose is no longer required in this new approach. The estimated hole pose can be deduced from DLVS and input to the RL policy. This improvement is significant as in real-world robotic assembly tasks, the pose of the part to be mated is normally unknown.

In future work, the DLVS model's generalization capability to different shapes can be evaluated without retraining. Optimal numbers of visual servoing iterations can also be found for different magnitudes of initial pose errors to boost the proposed approach's usability in real-life assembly tasks.

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
