# OpenReview forum: "Integrating Force-based Manipulation Primitives with Deep Learning-based Visual Servoing for Robotic Assembly"
_ICRA.org/2022/Workshop/Contact-Rich — ICRA 2022 Workshop: RL for Manipulation Poster_

### Official Review · Reviewer_dRWz · 2022-05-05
**Review of "Integrating Force-based Manipulation Primitives with Deep Learning-based Visual Servoing for Robotic Assembly"**

**Rating:** 7
**Confidence:** 4

**Review:**

Overall: This paper presents a method that combines deep learning-based visual servoing (DLVS) estimation and primitive-based RL policy for peg-in-hole tasks. Visual servoing is applied in the alignment phase while primitive-based policy is for insertion. Experimental results show the proposed method outperforms two baselines, as well as able to generalize to different workspaces.

Strengths:
The paper is clearly written and easy to read.
The proposed approach is a natural combination of [1] and [2]. The experimental results successfully demonstrate the effectiveness of the proposed approach (outperforming two baselines w.r.t. both insertion success rate and average completion time).
Generalization of visual servoing method is also shown in the experiments.

Weakness:
I wonder why the performance of the two baselines are so bad in terms of success rate. According to the result in Table 1, the visual servoing achieves estimation error less than 0.3mm in translation and 0.3 degree in orientation. In that circumstance, why would the visual servoing estimation with  pure compliance insertion only achieves 6% success rate. Also the RL policy only has 36% success rate. I doubt if the baseline methods are properly set.


[1] N. Vuong, H. Pham, and Q.-C. Pham, “Learning sequences of manipulation primitives for robotic assembly,” in 2021 IEEE International Conference on Robotics and Automation (ICRA), 2021, pp. 4086–4092.
[2] C. Yu, Z. Cai, H. Pham, and Q.-C. Pham, “Siamese convolutional neural network for sub-millimeter-accurate camera pose estimation and visual servoing,” in 2019 IEEE/RSJ International Conference on Intelligent Robots and Systems (IROS), 2019, pp. 935–941.

---

### Official Review · Reviewer_imGP · 2022-05-12
**Force-based Manipulation Primitives - Review**

**Rating:** 8
**Confidence:** 4

**Review:**

This work considers the task of peg-in-hole insertion using deep-learning based visual servoing. They perform both and insertion on a square peg randomly offset at initialization from the hole in both position and orientation. To complete this task, they use a reinforcement learning policy that samples manipulation primitives to control the end effector when in free-space and in-contact with the hole block. As baselines, they chose to compare their RL policy with a pure compliance-based insertion approach, and an RL-based policy without performing alignment first.

They collect a dataset of 4000 image samples of randomly sampled poses from a cylinder above the hole. Using this, they train the DLVS model by inputting a pair of images at two poses to a network and outputting the relative transformation between both of those poses. In their experiments, their insertion method performs both insertion and alignment better than the relative baselines. While they show their method generalizes within the distribution of sampled poses, they perform minimal tests outside of the sampling range (only 2), making it unclear how well their method performs on out of distribution samples. Overall, this paper could explore section E more thoroughly by including more test cases, in addition to adding more perturbations to the initial and final pose of the peg to show their method's robustness.